

# Chemical de-staining and the delta correction for blue intensity measurements of stained lake subfossil trees

Feng Wang[1, 2*], Dominique Arseneault[1, 2], Étienne Boucher[3], Shulong Yu[4], Steeven Ouellet[1], Gwenaëlle Chaillou[5], Ann Delwaide[6], Lily Wang[7]

[1]Département de Biologie, Chimie et Géographie, Université du Québec à Rimouski, Rimouski, G5L 3A1, Canada

[2]Centre d'études nordiques, Université du Québec à Rimouski, Rimouski, G5L 3A1, Canada

[3]Département de Géographie, GEOTOP and Centre d'études nordiques, Université du Québec à Montréal, Montréal, H3A 0B9, Canada

[4]Key Laboratory of Tree-ring Physical and Chemical Research and Xinjiang Laboratory of Tree-ring Ecology, Institute of Desert Meteorology, China Meteorological Administration, Urumqi, 830002, China

[5]Canada Research Chair in geochemistry of coastal hydrogeosystems, Québec-Océan, UQAR/ISMER, Rimouski, G5L 3A1, Canada

[6]Département de Géographie, Université Laval, Quebec City, G1V 0A6, Canada

[7]Institute of Geographic Sciences and Natural Resources Research, Chinese Academy of Science, Beijing, 100101, China

*Correspondence to*: Feng Wang (feng.wang@uqar.ca)



**Abstract.** The stain of wood samples from lake subfossil trees (LSTs) is challenging the wide application of the blue intensity (BI) technique for millennial dendroclimatic reconstructions. In this study, we used seven chemical de-staining reagents to

treat samples of subfossil black spruce (*Picea mariana* (Mill.) B.S.P.)) trees from two lakes in the eastern Canadian boreal forest. We subsequently compared latewood BI (LBI) and delta BI (DBI) time series along with conventional maximum latewood density (MXD) measured from the stained and de-stained samples. Results show that the stain of our samples is most likely caused by post-sampling oxidation of dissolved ferrous iron in lake sediments that penetrated into wood. Three reagents (ascorbic acid, sodium ascorbate and sodium dithionite all mixed with ethylenediaminetetraacetic acid) could remove >90%

of Fe. However, even for the best chemical protocol, a discrepancy of about +2°C compared to MXD data remains in the LBI-based temperature reconstruction due to incomplete de-staining. On the contrary, the simple mathematical delta correction, DBI is unaffected by Fe stain and shows very similar results compared to MXD data (r>0.82) from annual to centennial timescales over the past ~360 years. This study underlines the difficulty of completely de-staining lake subfossil samples, while confirming the robustness of the DBI approach. DBI data measured from stained LSTs can be used to perform robust

millennial temperature reconstructions.

## 1. Introduction

The recently developed blue intensity (BI) technique is potentially a valuable alternative to more expensive maximum latewood density (MXD) data for dendroclimatology (Björklund et al., 2019; McCarroll et al., 2002). MXD is the most suitable tree-ring parameter for summer temperature reconstructions in northern and high-altitude regions (Esper et al., 2014;

Frank and Esper, 2005). However, compared to less climate-sensitive ring-width data, millennial MXD series have been much less frequently developed worldwide, mainly due to the expensive X-ray densitometric facilities (Anchukaitis et al., 2017; St. George and Esper, 2019; Wilson et al., 2016). In contrast, BI is a cheap measure of blue light reflectance of tree-ring images generated using high-resolution flatbed scanners and image analysis software (Rydval et al., 2014). Several studies reported excellent coherences between latewood BI (LBI) and MXD data measured from living-tree materials of

several coniferous species across the northern hemisphere (Campbell et al., 2007; Kaczka et al., 2018; Österreicher et al., 2015; Rydval et al., 2014; Wilson et al., 2014), which encouraged researchers to further develop long BI series for climate reconstructions (Rydval et al., 2017; Wilson et al., 2019).

However, the BI technique is also facing challenges due to various color issues of wood materials. The best-known issue is the sapwood-heartwood color difference of several tree species such as pine and larch, leading to darker heartwood than sapwood (Björklund et al., 2014; Rydval et al., 2014; Wilson et al., 2019). Some wood types may also be discolored by

decay or weathering, a common issue with dead trees and historical materials (Wilson et al., 2014). Another less documented color issue happens with lake subfossil trees (LSTs), which often have darker wood than living trees (Wilson et al., 2019). LST is a very interesting source of material to extend tree-ring chronologies from centuries to several millennia and can greatly improve replications and robustness of reconstructions, especially in regions where only short-lived tree species



occur (Arseneault et al., 2013; Grudd et al., 2002). All the above color issues may potentially alter the measurement of BI
data (e.g. LBI) and introduce biases in BI-based climate reconstructions, particularly for the low-frequency domain
(Björklund et al., 2014; Wilson et al., 2019). Therefore, it is critical to develop unbiased BI data from a variety of wood
materials, in particular from LSTs, to help the promising BI technique to be widely applied in future dendroclimatic
reconstructions.

Some solutions have been proposed to overcome the sapwood-heartwood color issue. Sheppard and Wiedenhoeft
(2007) used hydrogen peroxide to bleach wood colors and attenuate the sapwood-heartwood difference, although an earlier
study claimed that chemical bleaching likely degraded the climate signal (Sheppard, 1999). A later and seemingly more
promising approach consists in a simple delta mathematical correction computed as the BI difference between the earlywood
and latewood, (i.e. the delta BI; hereafter DBI) of each tree ring. DBI is suitable to recover low-frequency (decadal–

centennial) variations distorted by the sapwood-heartwood color issues (Björklund et al., 2014, 2015). Only a few studies
attempted to use BI data measured from LSTs to develop long-term temperature reconstructions, but the staining issue was
not addressed and these BI series were not directly compared to MXD data. For example, Rydval et al. (2017) combined the
high-frequency temperature signals extracted from LBI with the low frequency from ring-width data of LSTs in order to
reconstruct temperatures, leaving the low-frequency domain unimproved by the more temperature-sensitive BI data.

In this study, we explore the potential of generating unbiased BI series from stained black spruce LSTs from two
eastern Canadian lakes (Fig. 1a). More specifically, we compare chemical de-staining with the DBI approach as well as with
conventional MXD data. The following hypotheses were formulated: (1) the stain of LSTs is mainly caused by oxidation of
Fe (Hyacinthe et al., 2006; Kostka and Luther, 1994; Pelé et al., 2015; Zhang and Xi, 2003), which can be removed using
some anti-oxidant reagents; (2) chemical de-staining and DBI can greatly improve BI-based temperature reconstructions,

making them comparable to MXD data. We first treated thin wood laths of stained LSTs using seven potential chemical
reagents and quantified the proportion of Fe extracted by chemicals. Subsequently, we performed dendroclimatic
assessments by comparing LBI and DBI chronologies of de-stained and stained LST samples with MXD data over the past
~360 years.

## 2. Materials and methods

### 2.1 Study sites and staining issue

The two studied lakes (L20 and L105) are located approximately 450km apart in the eastern Canadian boreal forest of
the Quebec-Labrador Peninsula (Fig. 1a). The climate of this region is characterized by short, mild summers and long, cold
winters (Environment Canada, 2020). Regional forests are strongly dominated by black spruce, mixed with balsam fir (*Abies
balsamea* L.) and eastern larch (*Larix laricina* (Du Roi) K. koch) (Payette, 1993). Lakes are extremely abundant and cover

up to about 25% of the landscape. Numerous black spruce trees in the lakeshore forests thus become LSTs once fallen in
water and eventually buried in lake sediments (Arseneault et al., 2013; Gennaretti et al., 2014a, 2014b). Black spruce LSTs



from L20 (54.56º N, 71.24º W) were previously included in a millennial temperature reconstruction (Gennaretti et al., 2014c). L105 (50.81º N, 67.80º W) is a new lake where more than one thousand black spruce LSTs were extracted, and many of them were successfully cross-dated to develop a millennial ring-width chronology (unpublished data).

LSTs in the eastern Canadian boreal forest are frequently stained to various blue-gray intensities (Fig. S1). When dry, these stains correspond well to the Munsell Soil Color Chart 2009-5Y-Chroma I. In total, 78% and 79% of the cross-dated LSTs at L20 and L105, were stained to some degrees of gray, respectively (Fig. 1b, c). Very few LSTs display additional colors, for example due to fungi invasion. Stained LSTs are distributed throughout the timespan of the two millennial chronologies with increasing proportions back in time, particularly before year 1800 CE (Fig. 1b, c), suggesting that staining
issues are unavoidable when using BI measurements from this material.

## 2.2 De-staining experiments and chemical analyses

We selected stem cross sections from five (evenly and heavily) stained LSTs and five unstained lakeshore living trees from each of the two studied lakes. Using a twin-blade saw (DendroCut, Walesch Electronic), we transversally cut sixteen 1mm-thick laths along the radii of each subfossil tree (total of 160 laths), along with 2 laths from each
living tree (total of 20 laths). All laths were pretreated using 95% ethanol in Soxhlet extractors for 48 hours to remove resins and then air-dried and weighted. 16 pretreated laths from each LST were divided into eight pairs. Seven pairs were immersed in 50 mL of one of the seven chemical solutions (Table 1) in Falcon® 50mL tubes (see an example in Fig. 2), then placed on an electronic shaker (SK-600, Montreal Biotech Inc.) with a speed of 133 rpm at room temperature (c.a. 20 ºC) for 24 hours (MixC) and 48 hours (other reagents). Treated laths were rinsed 4–5 times,
immersed in de-ionized water for 2 hours to remove dissolved elements absorbed by wood tissues, and then air-dried for subsequent analysis. The eighth pairs of subfossil laths and all the living-tree laths were not treated with de-staining reagents and considered as control samples. Design of the experiments is illustrated in Fig. S3.

During de-staining treatments, we sequentially sampled reaction solutions to construct temporal Fe dissolution curves (the most abundant metal element detected in preliminary tests) for MixA, MixB and MixC, the most effective
de-staining reagents (see results below). 1mL of solution was sampled at 0, 1, 3, 6, 12 and 24 hours for MixA and MixB, and at 0, 0.5, 1, 3, 6 and 12 hours for MixC (Fig. 2). 1mL extracted solutions were then diluted in 5mL of 5% (v/v) hydrochloric acid in Falcon® 15mL tubes in order to avoid Fe(II) precipitation prior to the chemical analysis of Fe concentrations. After de-staining, one lath of each laths pair was used to quantify the amount of residual Fe and was digested using 5mL nitric acid and 1mL hydrogen peroxide in a MARS-Xpress microwave digestion system (CEM
Corporation) at 150 ºC for 30 minutes (Fig. 2). Digested solutions were diluted to 25mL in volumetric flasks using de-ionized water. Fe concentrations were measured using the MP-AES (detection limit for dissolved Fe is ~4.6 ppb) and data were adjusted to "milligram of Fe per gram of wood" according to the dilution and weight of corresponding wood



lath. Fe concentrations in this study represent the total of ferrous and ferric Fe which cannot be distinguished by MP-AES.

The second lath of each pair was air-dried, finely sanded (to 1000 grits) and scanned using the SilverFast 8.0 software (LaserSoft Imaging) and an Epson V800 flatbed scanner. In order to obtain optimal calibration results, sanded laths were scanned to RGB images of 3200 dpi along with a color IT8.7/2 calibration target (LaserSoft Imaging). The scanner was covered by a black plastic box to avoid inferences of external light. It should be noted that the actual image resolution is approximately 2580 dpi (horizontal) by 1825 dpi (vertical) according to the USAF-1951 resolution

target. Wood RGB intensities (definition 1–3 in Table 2), were then measured using the CooRecorder 8.1 software (Cybis Dendrochronology). RGB values were subtracted from a value of 256 such that the inverted data are consistent to real-world observations, i.e. lower RGB values correspond to lighter colors. RGB intensities were compared among treatments (seven treatments plus two controls) to assess the efficiency of the de-staining reagents.

## 2.3 Chemical treatments for dendroclimatic assessment

In order to perform dendroclimatic assessments of the most effective de-staining treatments, we selected 57 trees of different types (stained and unstained LSTs as well as living trees) cross-dated after the year 1600 CE from L20 and L105 (28 and 29 trees, respectively; Fig. S4). We cut eight 1mm-thick laths from two radii of each tree to acquire four pairs of laths per tree (Fig. S5). Laths were pretreated using 95% ethanol in Soxhlet extractors for 48 hours. We then treated three pairs of laths per tree using the MixA, MixB and MixC, respectively, while keeping the fourth lath as an untreated control.

The conditions of de-staining treatments were the same as explained above, except that the treatments lasted for six hours, which is the time required for optimal de-staining according to the Fe dissolution curves (Fig. S2). Treated laths were then air-dried for LBI, DBI and MXD measurements.

## 2.4 Tree-ring data and chronology development

Measured wood and tree-ring parameters are explained in Table 2. Wood laths for dendrochronological assessments

were firstly X-rayed to generate MXD data prior to being sanded and scanned for BI measurements. X-ray densitometry experiments were conducted in a controlled environment with relative humidity of 50% and room temperature of 20ºC. X-ray films were developed using the DendroXray2 system (Walesch Electronic) and MXD series were measured using the Dendro2003 system (Walesch Electronic). LBI and DBI were measured using the same procedure as the measurement of wood RGB intensities. We subtracted the raw LBI values from a value of 256 in order to make the LBI positively correlated

with MXD data, according to Rydval et al. (2014) and Wilson et al. (2019). Before data analysis, LBI, DBI and MXD data were averaged by tree (i.e. by pair of laths) for each treatment (MixA, MixB, MixC, Control).

We used regional curve standardization (RCS) to remove the biological trends from tree-ring series in order to retain low-frequency (decadal–centennial) climatic variations (Briffa and Melvin, 2011; Helama et al., 2017). Age-dependent



spline with an initial stiffness of 2 years was used to estimate the regional curve. Standardized tree-ring series were computed as ratios between raw data and the smoothed regional curve. In total, we standardized 24 groups of tree-ring data by site (L105 and L20), parameter (LBI, DBI and MXD) and treatment (MixA, MixB, MixC and Control). We excluded data from chemically treated, unstained trees (unstained LSTs plus living trees) because we found BI data of unstained trees tend to be altered by reagents, particularly the MixC (Fig. S8). Consequently, we pooled the data of stained LSTs that were chemically treated, plus the data of the untreated, unstained trees for each standardization. In addition, living-tree data of L20 after the year 1950 CE were excluded as BI diverged from MXD data (Fig. S4 and S7), likely due to the sapwood-heartwood color issue and unhealthy tree growth (see discussions below).

Regional chronologies were generated by pooling standardized series from both sites using the Tukey's bi-weight robust mean for each tree-ring parameter and treatment. This approach was selected because of the limited tree replication per site (Fig. S2). Regional MXD chronologies from the four treatments were similar and thus averaged to one reference chronology. All regional chronologies were truncated at 1655 CE to ensure a minimum replication of five trees (Fig. S2c).

### 2.5 Temperature reconstruction

We performed temperature reconstructions using the regional LBI, DBI and MXD chronologies to further quantify the influence of the de-staining protocols. Instrumental summer (May to August) temperature data were obtained from the CRU TS 4.02 0.5° gridded monthly mean temperature dataset (Harris et al., 2014) and averaged from the four grid cells closest to each lake in order to generate a regional temperature target. Reconstructions were based on the scaling method (Esper et al., 2005; Rydval et al., 2017), which forces chronologies to possess the same mean value and standard deviation as the temperature target over the 1901–2015 time period.

### 2.6 Data analysis

Data were analyzed using the R program (R Core Team, 2018). We conducted linear regressions between wood RGB intensities and logarithmic residual Fe of both treated and untreated LSTs in order to determine the roles of Fe on the staining issue. For chronology assessments, we generated several high-pass and low-pass LBI, DBI and MXD series using the Butterworth filter available in the "dplR" R package (Bunn, 2008). Pearson correlation coefficients were used to assess the degrees of coherence among all the time series. Reconstruction performances were assessed following a regression-based calibration-verification procedure using the "treeclim" R package (Zang and Biondi, 2015). Since our chronologies showed higher replication during 1901–1960 (Fig. S4), this time period was used for calibration, while 1961–2015 period was used for verification.



## 3. Results

### 3.1 Effects of chemical de-staining

LSTs displayed a variety of color changes after the seven de-staining treatments (Fig. 3a). NaAsc resulted in very similar colors as the untreated stained samples, representing the weakest de-staining effect. Conversely, MixA, MixB and MixC showed dramatic effects and almost completely removed the gray stain. MixC was the most effective de-staining solution based on wood RGB intensities, although the resultant colors still slightly differed from the living-tree standards (Fig. 3a, b). BI is less variable with varying degrees of post-treatment stains in comparison to red and green intensities (Fig. 3b), suggesting a potentially weaker influence of wood stain on the BI data.

### 3.2 Stains versus iron

Chemical analyses showed strong links between Fe concentrations and color intensities of wood, especially for green and red ntensities (Fig. 3a, c). Total Fe concentration was highest for untreated stained LSTs and near zero for living trees (Fig. 3c). MixA, MixB and MixC could remove 94.1%, 92.5% and 96.2% of Fe relative to the amount measured in untreated stained LSTs, respectively. Although Fe dissolution curves stabilized after 6–12 hours (Fig. S2), minor quantities of residual Fe at the end of the treatments (24-48 hours) indicated that all de-staining reactions are incomplete. Significant ($p<0.001$) linear relationships existed between the log of residual Fe and post-treatment color (RGB) intensities of earlywood and latewood treated (Fig. 3d, e). However, such linearity markedly weakens for delta RGB intensities, especially for delta BI ($p=0.087$, Fig 3f), indicating that DBI of LSTs is insignificantly affected by the staining issue (Note that wood delta BI and DBI are not exactly the same as explained in definition No. 3 and 5 Table 2, respectively).

### 3.3 Comparison of LBI and DBI against MXD chronology

LBI chronologies of the four retained treatments (MixA, MixB, MixC plus Control) diverged relative to the reference MXD chronology (Fig. 4a) prior to year 1900 CE when stained LSTs dominate the chronologies (Fig. S4c). Correlation analyses showed that coherences between LBI and MXD chronologies are only robust for the 10-year high-pass filtered data ($r>0.89$), and decreased at longer timescales (Fig. 4c). In contrast, DBI chronologies are very similar to the reference MXD chronology for all treatments (Fig. 4b). Correlations with MXD data are strong and stable among all frequencies tested ($r>0.82$) (Fig. 4d). In addition, little differences were found between untreated control DBI series and chemically treated DBI data.



## 4. Discussion

### 4.1 Causes of stain

Significant relationships between post-treatment wood RGB intensities and residual Fe (Fig. 3), along with the rapid post-sampling staining (Fig. 5b), support our hypothesis that oxidation of Fe is a major cause of stain in our lake subfossil material. Fe is abundant in natural aquatic systems as dissolved and particulate fractions (Bortleson and Lee, 1974; Davison, 1993; Nürnberg and Dillon, 1993). Briefly, dissolved Fe, mainly in Fe(II) state, is reduced and mobilized in porewater in anoxic sediments. Dissolved Fe can migrate upward to oxic bottom water to form particulate Fe(III)-oxides (Davison, 1993;

Davison et al., 1982). This cycle results in much higher concentrations of dissolved Fe in anoxic sediments compared to oxygenated freshwaters (Zaw and Chiswell, 1999). Soluble forms of Fe in anoxic sediments can readily penetrate into buried wood tissues. When buried LSTs are extracted from lakes, cut and exposed to air, dissolved Fe is rapidly oxidized to colored Fe-oxides (oxyhydroxides, hydroxides and more crystalline Fe(III)-oxides) which may combine to wood (Pelé et al., 2015). This process is also supported by the fact that fresh cuts from buried portions were heavily stained while exposed (but

submerged) portions of the same LST were not (Fig. 5a). Photo-oxidation is assumed to be less likely, yet not improbable, because the stain contaminated both surface and inner portions of LSTs.

      Furthermore, we found that amorphous and crystalline Fe-oxides are likely produced during the oxidation of dissolved Fe. About 63.4% of Fe is removed by the NaAsc reagent, although post-treatment colors of LSTs only slightly lightened (Fig. 3a, c). The neutral NaAsc, similar to buffered ascorbates, only extracts the most reactive amorphous Fe-oxides

(Anschutz et al., 2005; Hyacinthe et al., 2006; Kostka and Luther, 1994). The less reactive crystalline phases can be removed by EDTA, HAsc, MixA, MixB and MixC, each of which removed at least 25% more Fe than NaAsc. In fact, EDTA, ascorbic acid and sodium dithionite, which are the active chemicals in these solutions, are known as useful extractants of both crystalline and amorphous Fe-oxides (Borggaard, 1982; Hyacinthe et al., 2006; Kostka and Luther, 1994; Tessier et al., 1979). The notable de-staining effect of these solutions (Fig. 4a) also implies that crystalline Fe-oxides are more color-

reflective than amorphous ones. In our experiment, HAc extracted less Fe than NaAsc but with a better de-staining effect (Fig. 3a, c). A probable explanation is that acid-soluble Fe-oxides extracted by HAc (Chester and Hughes, 1969; Gupta and Chen, 1975; Tessier et al., 1979) are in amorphous and crystalline phases which have stronger color reflectivity, whereas NaAsc only extracted less color-reflective amorphous Fe-oxides.

      Other metal elements most likely have a negligible staining effect on LSTs compared to Fe. Our preliminary analyses

demonstrated that, among 15 potential metal elements (including iron, manganese, chromium, cobalt, copper, lead and etc.), Fe was the only element present at high concentrations. Although manganese is relatively abundant in the samples (several times higher than other metals), its concentration is still approximately 20 times lower than Fe. In addition, we did not detect any copper and lead from our samples. On the other hand, Fe complexes bound to sulfur and phosphorus might also be responsible for the staining of LSTs along with Fe-oxides. However, the MP-AES instrument used in this study is not

sensitive to detect sulfur and phosphorus.



### 4.2 Chemical de-staining versus delta correction

Divergent trends of the three chemically treated LBI chronologies (MixA, MixB, MixC) compared to the reference MXD chronology demonstrate that none of the de-staining treatments can generate satisfactory and robust LBI data (Fig. 4a). Although MixC is the most effective protocol with very few residual Fe (<5% relative to the stained LSTs), the corresponding LBI chronology still displays a significant long-term bias before 1830 CE (Fig. 4a), leading to a discrepancy of about +2 ºC compared to the MXD-based reconstruction (Fig. 6a). If no chemical treatment had been applied, such temperature discrepancy would be amplified to about +4 ºC (not shown). These errors are caused by the extreme color sensitivity of LBI values as direct measures of blue light reflectance. Therefore, any stain contributing to the wood color will strongly contaminate the LBI data, especially in the low-frequency domain (Björklund et al., 2014). These results discourage the use of LBI in the presence of stained subfossils woods. By contrast, high-frequency variability of LBI data seems unaffected by the Fe stain (Fig. 4b). Coherence of high-pass filtered MXD series with LBI data is slightly higher than with DBI data measured from untreated stained samples (Fig. 4b, d), however, such results are based on a relatively small replication.

Unlike LBI data, DBI is unaffected by the Fe stain from annual to centennial timescales, which is shown by the high and stable coherence between DBI and MXD chronologies (Fig. 4b, d) and the non-significant linear relationships between log of residual Fe concentrations and delta BI data (Fig. 3f). The linearity between DBI and MXD remained almost unaffected by de-staining treatments, for example the most efficient MixC protocol (Fig. S9). Furthermore, compared to the Control DBI chronology, no chemical treatments substantially improved the correlation of DBI with MXD data (Fig. 4d), resulting in nearly identical trends in the corresponding temperature reconstructions in comparison with MXD, except for some periods where tree replication is less than 10 (Fig. 6c, d). These evidences suggest that DBI, is not only excellent to resolve the sapwood-heartwood color biases, but is also efficient to resolve the low-frequency biases caused by the Fe stain in black spruce LSTs from the eastern Canadian boreal forest.

We observed that LBI as well as DBI diverged from MXD data after 1950 CE at L20 (Fig. S7). Divergence of LBI is due to slight color differences between heartwood and sapwood of selected living trees although this issue is generally not serious for black spruce compared to pine or larch species (Rydval et al., 2014; Sheppard, 1999; Yang, 2007). DBI is theoretically sufficient to solve the sapwood-heartwood color issue (Björklund et al., 2014) but DBI only recovered some negative trend (Fig. S7). A large proportion of unhealthy old living trees were collected from lakeshore forests at L20 and they often displayed declined ring widths compared to healthy trees sampled at other sites (not shown). DBI of L20 is likely influenced by the ring-width declines because DBI of black spruce is not only similar to MXD but also the ring-width data (Wang et al., in preparation). We thus speculate the divergence of DBI only reflects the site-specific unhealthy tree growth.

This study confirms the robustness of DBI data from stained black spruce LSTs. Yet, two points need be considered for future DBI-based climate reconstructions. Firstly, compared to MXD a higher tree replication is often needed for DBI data to obtain equivalent climatic signal (Rydval et al., 2014; Wilson et al., 2019). Thus, it is not surprising that the control DBI-





based reconstruction shows weaker verification statistics against instrumental temperature (Table S1) and some slight

instability during poorly replicated periods (Fig. 6c, d). When replication is above 15 trees during 1901–1960 (Fig. S4c), the calibration $r^2$ is similar for DBI and MXD data against temperature (Table S1). Secondly, Björklund et al. (2014, 2015) suggested some multi-centennial biases in DBI data due to the heterogeneous wood color. Although this phenomenon is not obvious in our case regardless of the Fe stain, our reconstruction only spanned the last three centuries and future attentions are needed to verify this potential bias.

**5. Conclusion**

        Our study indicates that the simple delta correction of differentiating latewood and earlywood BI values is more effective to resolve the staining biases of BI data from LSTs than the much more complex and time-consuming chemical de-staining protocols tested here. DBI of black spruce LSTs is unaffected by the Fe stain from annual to centennial timescales and allows robust temperature reconstructions similar to MXD data. Consequently, DBI from stained black spruce LSTs is a

promising proxy for developing millennial temperature reconstructions in the eastern Canadian boreal forests, a region with very few long MXD series (Wang et al., 2001). On the contrary, LBI is very color sensitive and appears problematic in retaining the low-frequency climatic signals.

        The chemical de-staining experiments, though not satisfactory regarding the robustness of LBI data, suggests the mechanisms of the staining issue through post-sampling chemical Fe oxidation. Since Fe is so abundant in the Earth's

systems, our results may be representative of much wider regions. On the other hand, excellent Fe extraction abilities (removal of >90% Fe) of three chemical mixtures also suggest that they, in particular the MixC, can be further used as Fe extraction protocols for waterlogged archeological artifacts which are also facing Fe-staining issues (Fors et al., 2014, 2012; Pelé et al., 2015; Zhang and Xi, 2003).

*Data and sample availability*. Instrumental temperature data is available at Climate Exploror: https://climexp.knmi.nl/start.cgi. Other data will be available online after acceptance of this article. The wood samples are stored at the University of Quebec in Rimouski.

*Author Contributions*. FW, DA, ÉB, SO and GC designed the experiments. SY, GC, AD and LW supported facilities and
guided the experiments. FW, SY, SO and GC performed the experiments. FW and DA analyzed the data. FW and DA prepared the original manuscript with inputs from ÉB, GC, SO and AD.

*Competing interests*. The authors declare that they have no conflict of interest

*Acknowledgments*. This work was supported by the "PERSISTENCE" project funded by the Natural Sciences and Engineering Research Council of Canada (the NSERC-DRC program), Hydro-Québec, Manitoba Hydro and the Ouranos



Consortium. FW was also funded by the China Scholarship Council. SY was supported by Key Laboratory Opening Subject of Xinjiang Uyghur Autonomous Region (2016D03005) and Basic Research Operating Expenses of the Central-level Non-profit Research Institutes (IDM201202). LW was supported by National Natural Science Foundation of China (41571094). We greatly thank Florent Vignola for his assistance in cutting wood blocks. Leila Jolicoeur, Nadège Trou-Kechout and Pauline Balducci helped during laboratory and fieldworks. We also thank professor Feng Chen and Tongwen Zhang for their permissions to perform tree-ring density measurements at Institute of Desert and Meteorology, China Meteorological Administration.



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



**Tables:**

**Table 1.** Basic chemical properties of the seven chemical solutions. All solutions were diluted using de-ionized water with no trace of Fe concentration. Concentrations are in v/v for HAc and in w/v for other solutions. EDTA: Ethylenediaminetetraacetic acid. Values of pH were estimated using pH test papers.

| Code | Chemical components | Chemical property | pH |
|------|---------------------|-------------------|-----|
| NaAsc | 2% sodium ascorbate | reduction | 7 |
| HAc | 2% acetic acid | acidity | 3 |
| EDTA | 2% disodium EDTA | chelation & acidity | 5 |
| HAsc | 2% ascorbic acid | reduction & acidity | 3 |
| MixA | 2% ascorbic acid + 2% disodium EDTA | reduction, chelation & acidity | 4 |
| MixB | 2% sodium ascorbate + 2% disodium EDTA | reduction & chelation | 7 |
| MixC | 2% sodium dithionite + 2% disodium EDTA | reduction, bleaching & chelation | 7 |

**Table 2.** Definitions of wood color intensities and tree-ring parameters used in this study. *: data were inverted by subtracting the raw data from a value of 256.

| No. | Parameter | Definition |
|-----|-----------|------------|
| 1 | earlywood & latewood RGB intensities* | Mean RGB intensities of entire earlywood & latewood averaged from one entire wood lath (see Fig. S6a). |
| 2 | wood RGB intensities* | Mean RGB intensities by averaging earlywood and latewood RGB intensities. |
| 3 | delta RGB intensities | Earlywood RGB intensities subtracted from corresponding latewood RGB intensities |
| 4 | LBI* | Mean blue intensity of 30% darkest pixels in latewood (Fig. S6b). |
| 5 | DBI | Raw LBI subtracted from raw earlywood BI, automatically derived from CooRecorder 8.0. |
| 6 | MXD | The maximum value of measured tree-ring latewood density. |





**Figures:**

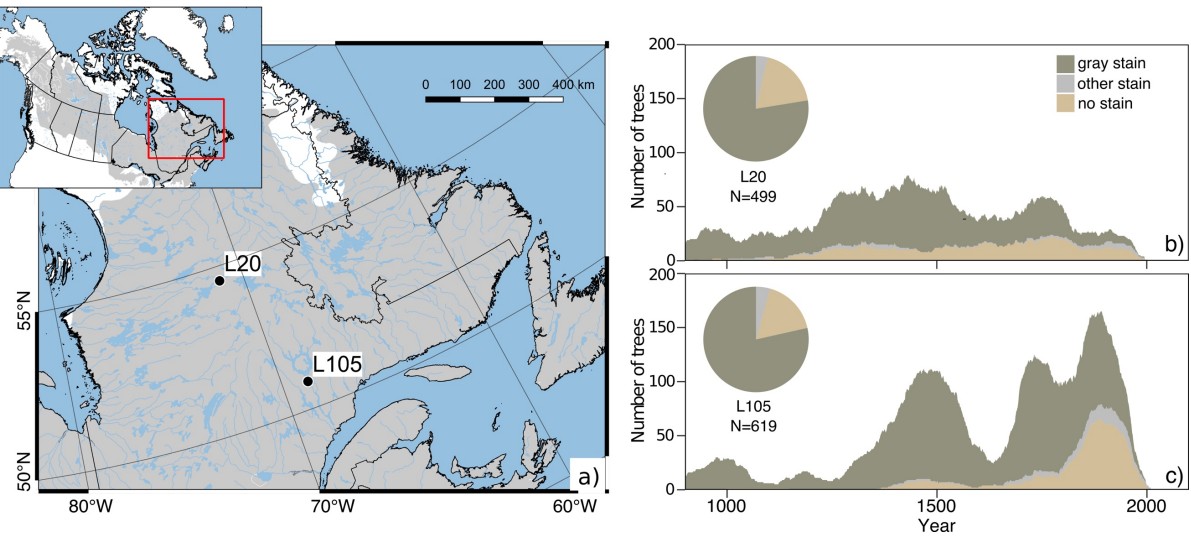

Figure 1. Location of the two studied lakes (a), and frequency of cross-dated LSTs according to staining at L20 (b) and L105 (c).
The gray shades in (a) corresponds to the distribution range of black spruce. The "other stain" category includes a variety of
additional colors such as red or dark brown.

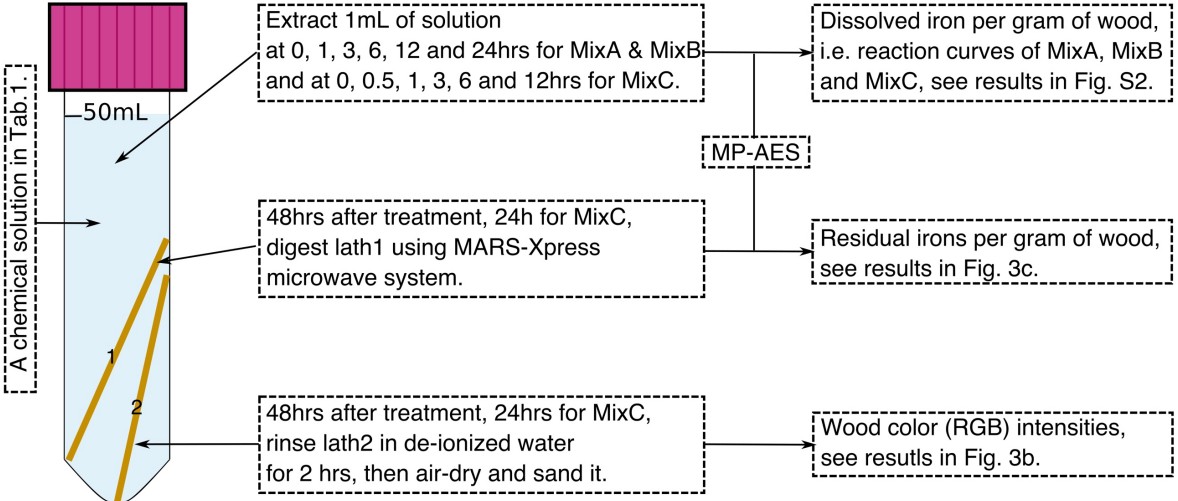

Figure 2. Diagram of one chemical de-staining experiment for one pair of wood laths from the same subfossil tree in a Falcon®
50mL tube.

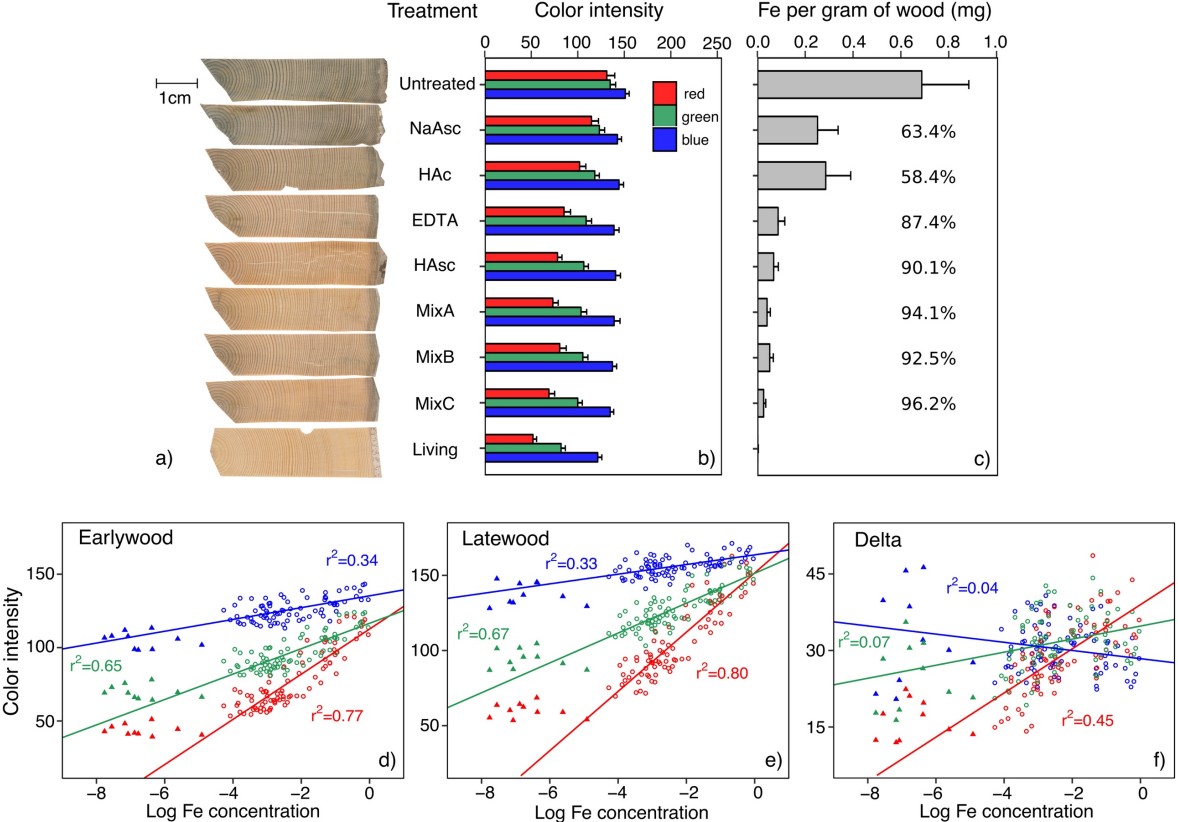

Figure 3. Residual iron (Fe) and wood RGB intensities (see definitions in Table 2) of LST laths treated with seven chemical reagents. (a) shows an example of one LST and one living tree sample (last row). The gray outer part of the example LST is discolored due to decay. (b) and (c) are mean wood RGB intensities and mean concentrations of residual Fe along with the corresponding standard deviations (error bars). Percentages in (c) refer to the Fe removed by de-staining treatments relative to the Fe concentrations of untreated stained LSTs. (d)–(f) show the linear regressions of earlywood, latewood and delta RGB intensities against the log of residual Fe. Regressions are based only on the LST data (circles). Living-tree data are plotted as triangles and are excluded from the regressions.

55



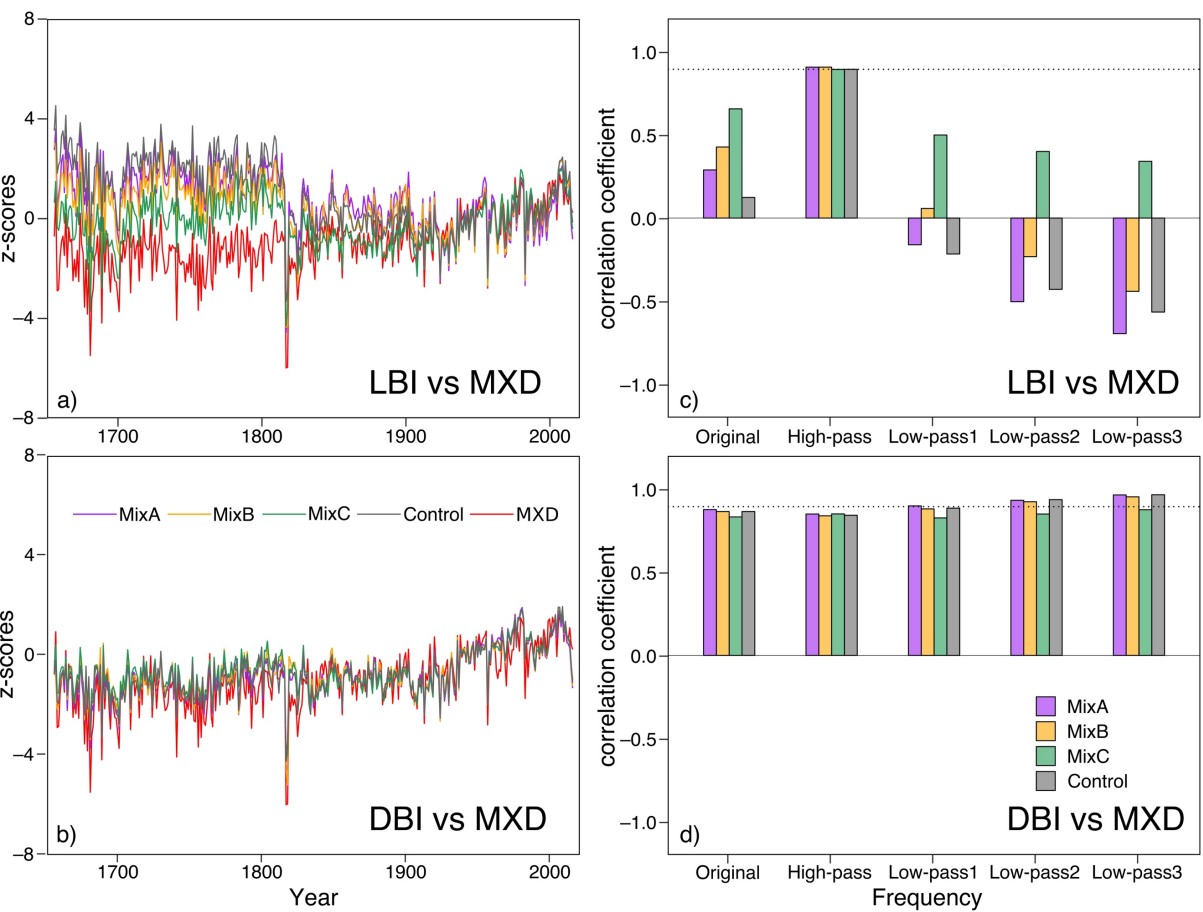

**Figure 4. Comparisons of LBI (a, c) and DBI chronologies (b, d) for the MixA, MixB, MixC and Control treatments against the reference MXD chronology. In (a) and (b), chronologies are transformed to z-scores relative to the 1901–2015 time period. (c) and (d) show correlations of LBI and DBI chronologies against MXD chronology at different timescales. Original: the original RCS standardized chronologies; High-pass: 10-year high-pass filtered series; Low-pass1, 2 and 3: 10-year, 50-year and 100-year low-pass filtered series. Dotted horizontal lines in the right panel show the correlation (*r*=0.89) between the high-pass filtered LBI and MXD chronology.**





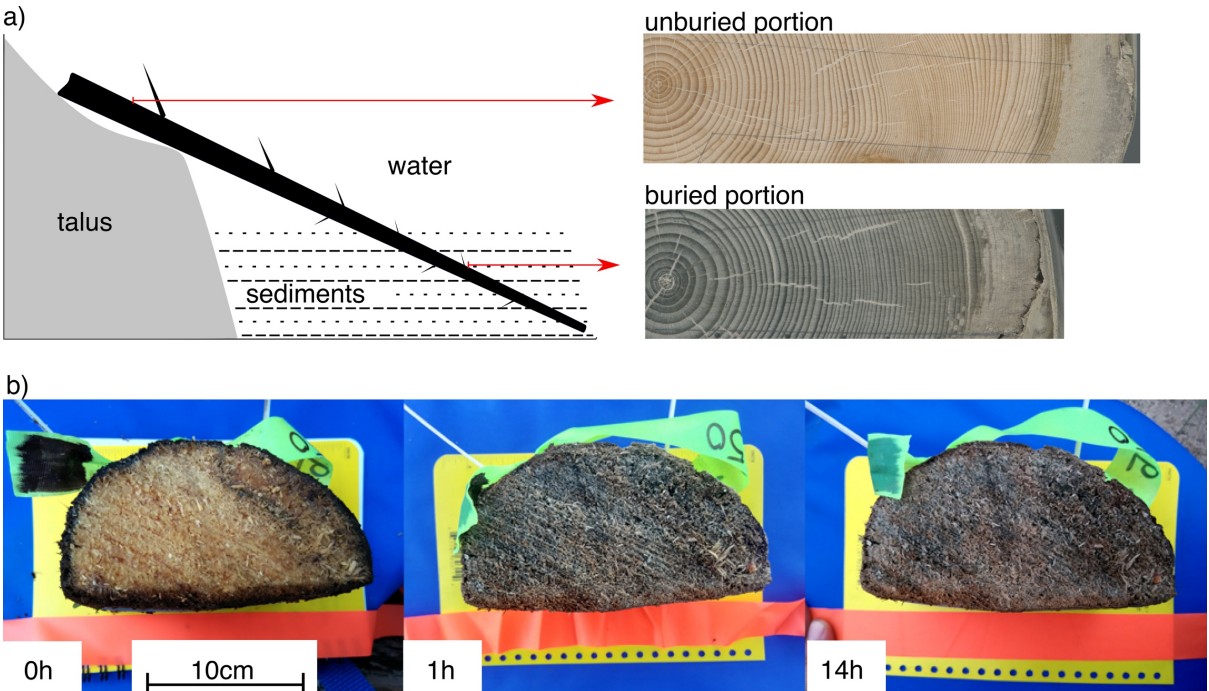

**Figure 5. Schema of different cross-sectional colors from the buried and exposed cross sections of the same partially buried tree (a), and field observations of cross-sectional color changes after a fresh disc was cut from a buried tree and exposed to air (b).**



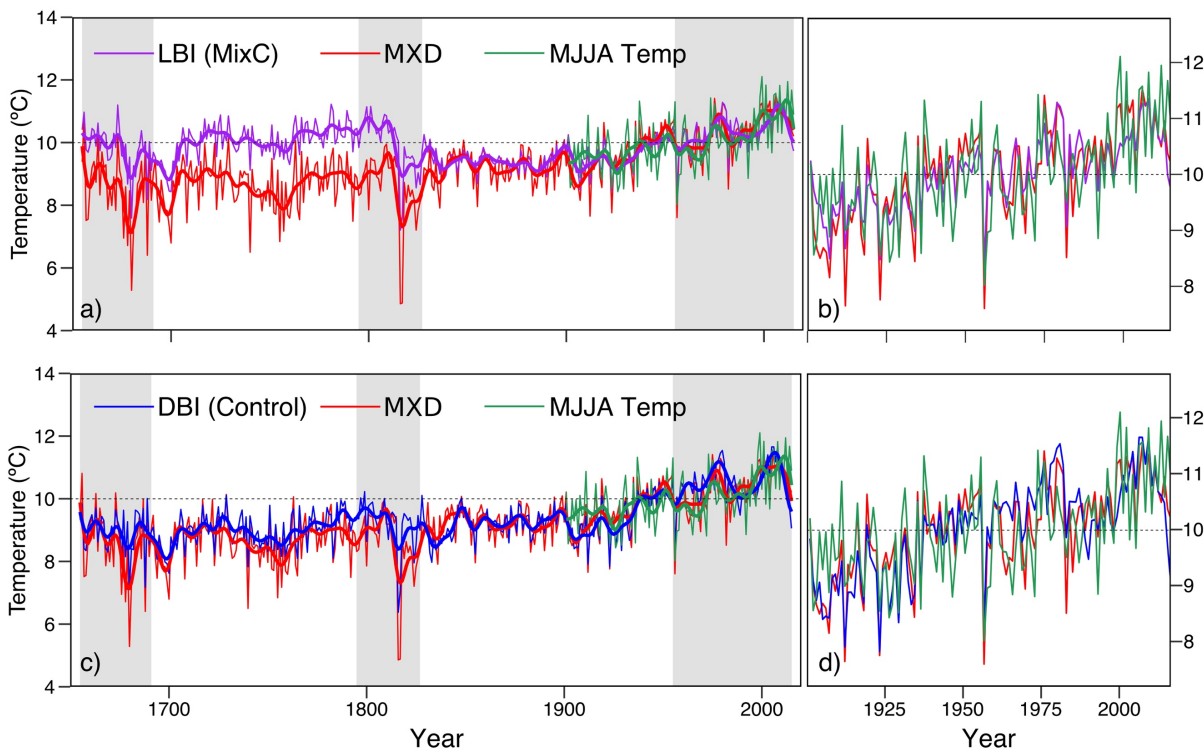

70

**Figure 6. Temperature reconstructions using the LBI chronology for the MixC protocol (purple) and DBI chronology for Control (blue) and the reference MXD chronology (red) for the 1655–2015 (a, c) and 1901–2015 (b, d) time periods. Thick smooth lines denote the 10-year low-pass filtered series. Vertical gray bars show the periods where tree replication is less than 10 (Fig. S4c).**