# Peer review of "Chemical de-staining and the delta correction for blue intensity measurements of stained lake subfossil trees"

_Biogeosciences, 2020_

## Referee Comment (RC1) · Milos Rydval (Referee) · 15 May 2020

General comments:

The manuscript addresses a highly topical issue related to the dendroclimatic utilization of reflected light (blue intensity) from lake subfossil wood material, specifically a discoloration bias related to staining, which is primarily attributed to Fe oxidation. The study investigates the application of a range of chemical treatment techniques in order to improve light reflectance data by reducing staining bias and also provides additional validation for the applicability of the delta BI correction procedure in the context of using samples affected by staining.

[Figure]

Overall, it is a nice, relevant and focused paper building on previous work on this topic. The study is designed and performed in a methodical manner and the manuscript is generally well organized and logically structured. Although I do not have any major comments on the methods used or analysis performed, the (mostly minor) comments detailed below will hopefully help to further refine this work. I would also recommend checking the manuscript to make minor language improvements (for example in order to clarify the meaning of certain statements) and I include specific suggestions (under minor comments) to indicate parts of the text where I believe most improvements could be made.

Specific comments:

L83: what is meant by 'new lake'? as in 'newly sampled lake' (i.e. data from samples from this lake were not analyzed / published before)? – please clarify

L106: Why were the treatment times different for MixC? Please explain briefly.

L117-118: Please add relevant reference(s) here.

L135-138: Please include information about the measurement resolution of the photo sensor used (i.e. the step size along the density measurement profile) – e.g. 10$\mu$m or variable?

L147-148: In what way were they affected – structurally, their color? If this is true for the unstained samples, could this play a role in affecting the properties of stained samples to some degree as well? Undoubtedly, the treatments lead to improvement, but it would also be worth discussing if there are (or could be) some undesirable effects as well that may perhaps limit the observed improvement?

L150: Was there any apparent difference in the sapwood / heartwood of the subfossil samples and if so, could that then potentially also have some effect on the results?

L152-153: Just to clarify - 'regional chronologies' here represents different parameters / treatment methods and not the two sites (since those were pooled together)? A slight

re-wording might help to ensure that this is clear.

L263: Slight reformulation is required here since the higher replication is needed to obtain a robust and representative (DBI) chronology rather than a chronology with an equivalent (or similarly strong) climatic signal to MXD (although it is true that these two things usually go hand in hand).

L264: At the same time, DBI appears to calibrate more weakly compared to LWB over the instrumental period. Could you suggest a possible explanation for this? Could you also add some moving window EPS statistics somewhere (e.g. in Figure 6) for the final chronologies used for the reconstruction in order to get a better idea which parts of the chronologies might be stronger / weaker.

L278-279: I think this sentence could be reformulated somewhat and expressed more clearly.

Table 2: It would be helpful to clarify a few things. To avoid any possible misinterpretation, it would help to specify in the caption that RGB refers to separate (R, G, and B) color channels rather than for example a full color RGB light values. Also, some more details should be included in relation to no. 3 and 5 to clarify the difference between these two terms – presumably delta RGB refers to all of the color channels whereas DBI only refers to the blue channel? Or is there some difference when it comes to how delta B (in RGB) was calculated compared to DBI? Unless I am mistaken, the settings for calculating DBI (i.e. how LBI and EBI are determined) in CooRecorder are adjustable – the main settings for this calculation should probably also be stated here.

Figure 1 caption: 'The gray shades in (a) correspond to . . .' / 'The gray shading in (a) corresponds to . . .'. Also, maybe simply use the term 'replication' instead of 'distribution range' here?

Figure 2: Certainly in the main text and maybe also here (e.g. in the figure caption) define MP-AES. For text in right-middle box consider: 'Residual iron content per gram

of wood ...'. Also, it may be more suitable to use past tense in the text boxes: e.g. 'lath1 digested using ...'; 'lath 2 rinsed in de-ionized water for 2 hrs, then air dried and sanded', etc.

Figure 3 caption: The description for (a) should make it clear that these are multiple laths from one LST - for example: '(a) shows examples of differently treated laths from one LST and one living tree sample (last row)'. When referring to panels (b) and (c), it would be clearer to write it using the following format: (b) description, and (c) description – (instead of (b), and (c) description)

Figure 6: Consider also adding some statistics in the panels such as the full period calibration. In the caption, maybe also specify the type of filter used.

Minor comments:

L36: Consider changing to e.g. '... high cost of X-ray densitometric equipment' ... unless this is meant to refer to the relatively high costs associated with processing samples at a 'facility' that is equipped to perform X-ray densitometry. In any case, a small edit is needed.

L37: It may be more accurate to state that the production of BI is 'relatively cheap' rather than 'cheap'.

L39: 'coherence' instead of 'coherences'

L43: Perhaps be a bit more specific here by changing 'color issues' for example to something like 'color inconsistencies' or 'inconsistent color properties'?

L44/45: As this is a property of the wood, 'leading to' is not really appropriate here – instead perhaps ' ... exhibiting darker heartwood than sapwood'

L47: 'occurs' instead of 'happens'

L48: 'LSTs are ...'?

L49: just 'replication'

L53: The second part of this sentence could be improved for example along the lines of '. . . to realize the potential of the promising BI technique . . .'

L58: 'consists of' instead of 'consists in'; the word 'delta' can be removed from this part of the sentence

L59: Consider changing 'DBI is suitable to recover . . .' for example to 'DBI suitably represents . . .' or 'DBI corrects for . . .'

L64: The last part of the sentence should be re-phrased e.g. '. . . without utilizing low frequency information of the more temperature-sensitive BI data' / '. . . without benefitting from / exploiting any potential improvements in the low frequency domain from the more temperature-sensitive BI data'

L69/70: Consider changing the last part of the sentence to something along the lines of '. . . to a standard comparable to MXD-based reconstructions'

L80/81: minor change needed here, e.g. '. . . LSTs after falling into the water and eventually becoming buried in lake sediments'

L88: 'fungal' rather than 'fungi'? Also, is there a better word that could replace 'invasion' here? Perhaps simply 'fungal discoloration'?

L89: 'before the year'

L94: Should this be 'radius' instead of 'radii' if laths are cut along a single radius, otherwise specify the number of radii

L96: This should probably be 'weighed' instead of 'weighted'

L102: I would recommend presenting the figures in the order that they are first mentioned in the text (Fig. S1 is first mentioned here while Fig. S2 is first mentioned in L131).

L108: This could probably be improved slightly – e.g. 'one lath from each pair of laths'?

L111: Please mention first that MP-AES stands for "microwave plasma-atomic emission spectrometer"

L112: '. . . weight of the corresponding . . .'

L113: The meaning of this sentence was not immediately clear to me. Consider the following minor revision for the sake of clarity: '. . . represent the combined total . . . cannot be separately distinguished by MP-AES'

L115: 'grit' rather than 'grits'

L118: 'interference' instead of 'inferences'

L122: I think this could be explained a bit better – in a more specific way. Also, 'consistent with' rather than 'consistent to'

L141: perhaps '(i.e. each lath pair)'?

L143: 'An age-dependent spline . . .'

L148/L150: Also in relation to Fig. S8 / S7, see earlier comment about sequential order of figures mentioned above.

L151: maybe 'poor (tree) health' would be better than 'unhealthy tree growth'

L154: 'averaged into'? Also, maybe consider showing the individual MXD chronologies in the SI.

L161: This could use a bit of re-wording.

L165: maybe slightly re-word – e.g. '. . . in order to assess the role of Fe in the staining issue'

L167: Perhaps just briefly specify the advantage of applying this specific filter in the context of high-pass / low-pass filtering.

L168: 'Performance of the reconstructions was assessed ...'

L170: '. . . while the 1961-2015 period . . .'

L178-179: Just as a general point, I wonder what the reason for this might be? Could it be related to the 'color properties' of the staining caused by Fe?

L182: 'intensities'

L186-187: '. . . of treated earlywood and latewood'?

L189: Please be more specific here and elaborate on what the difference is.

L191: Consider something along the lines of '. . . four treatments examined in more detail . . .'

L192: 'prior to 1900 CE' or 'prior to the year 1900 CE'

L193: 'coherence' rather than 'coherences'

L196: 'few differences' instead of 'little differences'

L203: 'Briefly' is probably not needed here

L208: I would recommend re-wording 'combine to wood' – 'combine with wood' or maybe 'bind to wood'?

L225: just 'etc.' instead of 'and etc.'

L228: maybe 'in our samples' rather than 'from our samples'?

L229-230: Minor edit needed here: e.g. 'is not sensitive to sulfur and phosphorous' / 'is not designed to detect sulfur and phosphorous' or 'is not sufficiently sensitive to detect . . .'?

L234: '. . . very little residual Fe . . .'

L236: probably replace 'such' with 'the'

L240: maybe something like '. . . when staining is present in subfossil wood' would be better

L242: 'a' should be removed

L247: something is missing here – e.g. '. . . for example as with the most efficient MixC protocol' or something similar

L250-251: 'This evidence suggests that DBI is not only an excellent solution to resolving ...'

L251: 'but also efficiently resolves ...'

L256-257: The last part of this sentence could be re-worded and expanded a bit to clarify what is meant by this.

L258: replace 'declined' with 'declining'

L259: 'decline' instead of 'declines'? Also, the reference Björklund et al., 2019 (Reviews of Geophysics) may also be relevant to this point.

L261: 'need to be' rather than 'need be'?

L268-269: '. . . further attention / investigation is needed . . .'

L281: should probably be '. . . used as part of Fe extraction protocols . . .'

L282: 'which also face / are also susceptible to / also suffer from'?

---

## Referee Comment (RC2) · Jesper Björklund (Referee) · 5 Jun 2020

The manuscript by Wang et al presents a very interesting sample material for temperature reconstructions and examine how to best utilize this in conjunction with the popular and affordable BI technique. The paper is foremost dedicated to a very novel and clever de-staining experiment which I thoroughly enjoyed and have the potential to be highly cited in future BI studies. The second component was a careful comparison of LBI, DBI and MXD from parallel X-ray measurements to evaluate the performance of the chemical de-staining and LBI and DBI parameters with MXD as reference. Although the authors conclude that the simple DBI was more successful in replicating the

low-frequency variance of the MXD, they have made some very important discoveries in terms of de-staining of relict wood material. The DBI parameter appears to be quite successful, but has documented problems as the authors also mention in the final sentences. Therefore, all tools available for de-staining prior to DBI transformation must be considered of great value. I congratulate the authors to a fine, and from what I can tell labor intensive, experiment and I consider the manuscript suitable for publication following minor revisions and clarifications. I also look forward to learn more about the planned follow-up manuscript.

Detailed comments:

L32 I would not say BI is recently developed anymore, it has been around almost 20 years now.

L32-33 The BI technique is an alternative to the X-ray technique in producing proxy parameters such as MXD.

L37 -38 Consider changing to something like: "In contrast, BI is more affordable because of the utilization of commercial flatbed scanners to generate images of reflected blue light analyzed in potentially affordable image analysis software..."

L38-42 Strange sentence, some of the studies encouraging more studies were made later than the encouraged studies. Work a bit more on this sentence and consider also these references: Björklund et al., 2014, 2015; Dolgova, 2016; Fuentes et al., 2018; Kaczka et al., 2017; McCarroll et al., 2013; Rydval, Gunnarson, et al., 2017.

L45 Should perhaps add something like: "..not accompanied by a similar difference in density.."

L83 newly exploited lake?

L84 millennium-long?

L96 What was the purpose of the weighing? Were the laths also weighed after

the chemical analysis? Could not find any more use of these measurements in the manuscript

L104 .., to identify the most effective. . .? Remove "(see results below)". The results are always be presented after the methods description..

L118 sensu Rydval et al., 2014?

L118-119 Great initiative

L121-122 Very strange statement. Real world observations? Do you mean: lower RGB values corresponds to lighter densities?

Section 2.2, 2.3 and 2.4 Consider re-structuring here. Perhaps one section for chemical de-staining description. One section for BI and X-ray data development and one section for chronology development for climate analysis, and sample average RGB data?

L139 Did you use the full RGB spectrum or only the blue spectrum? If the latter, it is consistent with the use of BI based parameters. Same comment in L165.

L145 N.B. residuals are most often used for density related parameters. This is not a major problem here since you compare results from BI and X-ray, but may be important in pure climate reconstructions.

L168 "coherence" can also be a type of statistical analysis, perhaps change to the more general term of "agreement", or simply not explain correlation since more or less the entire readership is familiar with this..

Figure S4 Spelling of replication

Figure s6 spelling of earlywood. It seems odd that the area of the 30% of the darkest pixels in the latewood are differently sized even though the latewood area is roughly the same (compare ring 4 and ring 5). Please check the definition you used and clarify why this is the case.

L169-171 Would be great to have running Rbar or EPS, to evaluate the difference between the different parameters. Perhaps this can explain why the DBI perform so badly in the post 1960 period compared to LBI and MXD. Both in terms of trend and correlation..

L182 spelling intensities

Figs. S7-S8 Would be interesting to also present the Earlywood measurements. Would be even more interesting if you also presented Delta density and Earlywood density.

It is puzzling why LBI and DBI has such similar trends in S7. Is there a HW/SW transition in these trees, if so why so weak in the earlywood? Are the rings in the post 1960 period very narrow? If so, I think that your measurement resolution is causing some problems here. Consider that the measurement resolution is affecting your latewood measurements more than your earlywood measurements. That is, your latewood BI is deflated because of adjacent contamination of earlywood BI. Ergo the delta BI will be artificially lowered and similar in trend to LBI.

Not completely relevant to your nice study, but could not resist :)

L196 check grammar

L208 combine to wood? Not clear, rephrase..

L241-242 This is not surprising. If you would calculate delta density and correlate with delta BI you would probably find equally high correlation as between LBI and MXD. This is not needed in revision, I am merely pointing this out.

L253-260 I think you are right that the narrow ring widths are causing the problem here, but I would not say it is a healthy versus unhealthy tree problem. It is a problem of measurement resolution (see comment above for fig s7). Healthy tree can also have narrow rings..

L262 yes interesting observation. Would be better underpinned if you also presented

the rbar for all the parameters.

Hope these comment can be helpful

Good luck!

---

## Author Comment (AC1) · 14 Jun 2020

We greatly appreciate the comments from Dr. Miloš Rydval and his kind line-by-line suggestions to improve the English expressions. Our responses are listed below:

General comments: The manuscript addresses a highly topical issue related to the dendroclimatic utilization of reflected light (blue intensity) from lake subfossil wood material, specifically a discoloration bias related to staining, which is primarily attributed to Fe oxidation. The study investigates the application of a range of chemical treatment techniques in order to improve light reflectance data by reducing staining bias and also provides additional validation for the applicability of the delta BI correction procedure

[Figure]

in the context of using samples affected by staining.

Overall, it is a nice, relevant and focused paper building on previous work on this topic. The study is designed and performed in a methodical manner and the manuscript is generally well organized and logically structured. Although I do not have any major comments on the methods used or analysis performed, the (mostly minor) comments detailed below will hopefully help to further refine this work. I would also recommend checking the manuscript to make minor language improvements (for example in order to clarify the meaning of certain statements) and I include specific suggestions (under minor comments) to indicate parts of the text where I believe most improvements could be made.

Specific comments:

L83: what is meant by 'new lake'? as in 'newly sampled lake' (i.e. data from samples from this lake were not analyzed / published before)? – please clarify

Response: We corrected "new lake" to "newly sampled lake".

L106: Why were the treatment times different for MixC? Please explain briefly.

Response: MixC was the last, but the most efficient and active reagent we used to treat samples. We had known that reactions of other (less active) reagents would terminate in 12–24hrs before using the MixC reagent. We thus used a shorter reaction time (24hrs) for MixC treatments, while we sampled the solutions of MixC more intensively (at 0.5hrs after treatments; Figure S2) in order to capture the change point of reaction (although finally this figure was only placed in the supplementary material). We added a sentence to explain the different time settings "Treatment and sampling times were set shorter for MixC since it was the most active reagent." in Lines 106–107 in the revised manuscript.

L117-118: Please add relevant reference(s) here.

Response: We added reference to (Rydval et al., 2014).

L135-138: Please include information about the measurement resolution of the photo sensor used (i.e. the step size along the density measurement profile) – e.g. $10\mu$m or variable?

Response: The resolution of the density profile is $10\mu$m. This information was added at Line138.

L147-148: In what way were they affected – structurally, their color? If this is true for the unstained samples, could this play a role in affecting the properties of stained samples to some degree as well? Undoubtedly, the treatments lead to improvement, but it would also be worth discussing if there are (or could be) some undesirable effects as well that may perhaps limit the observed improvement?

Response: The major effects are related to colors. As we can see from the unstained+living tree group on the right panel of Figure S7 (in the revised supplementary material), raw LBI, DBI, and MXD measurements were similar for MixA, MixB, and Control treatments, while MixC altered the LBI and DBI but not the MXD measurements. We think the bleaching effect of MixC is weak on the subfossil wood since DBI of stained LSTs did not diverge while DBI of living trees and unstained trees diverged. We added a few sentences at the end of the Result section 3.1 (Line 184–187) to discuss this effect. "MixC, the only reagent with bleaching function (Table 1), could have an additional bleaching effect on the wood, resulting in smaller LBI and DBI values in living and unstained trees compared to untreated control (Fig. S7). However, DBI of the stained LSTs was only slightly modified by the MixC treatment (Fig. S7), indicating that the bleaching effect of MixC is weak for the stained samples."

L150: Was there any apparent difference in the sapwood / heartwood of the subfossil samples and if so, could that then potentially also have some effect on the results?

Response: The sapwood portion of subfossil samples is generally decayed due to their long stay in water/sediments. We thus did not observe such apparent sapwood-heartwood difference on the subfossil samples. In addition, black spruce in general

does not show apparent color difference between the sapwood and heartwood. Living-tree samples of L20 may represent a special case where this color issue occurred.

L152-153: Just to clarify - 'regional chronologies' here represents different parameters / treatment methods and not the two sites (since those were pooled together)? A slight re-wording might help to ensure that this is clear.

Response: We changed the sentence to "Regional chronologies for each tree-ring parameter (LBI, DBI, and MXD) and treatment (MixA, MixB, MixC, and Control) were generated by pooling standardized series from both sites using the Tukey's bi-weight robust mean." in Lines 153–154.

L263: Slight reformulation is required here since the higher replication is needed to obtain a robust and representative (DBI) chronology rather than a chronology with an equivalent (or similarly strong) climatic signal to MXD (although it is true that these two things usually go hand in hand).

Response: We changed the sentence to "Firstly, a higher tree replication is often needed for DBI than MXD data, in order to obtain a robust chronology (Rydval et al., 2014; Wilson et al., 2019)." in Lines 270–271.

L264: At the same time, DBI appears to calibrate more weakly compared to LWB over the instrumental period. Could you suggest a possible explanation for this? Could you also add some moving window EPS statistics somewhere (e.g. in Figure 6) for the final chronologies used for the reconstruction in order to get a better idea which parts of the chronologies might be stronger / weaker.

Response: We added the moving EPS to Figure 6e in the revised manuscript (See Figure in the supplementary reply letter). We also discussed the more variable EPS of DBI in contrast to LBI and MXD at Line 273.

L278-279: I think this sentence could be reformulated somewhat and expressed more clearly.

Response: We changed the sentence to "The chemical de-staining experiments, though not satisfactory regarding the robustness of LBI data, suggest that the post-sampling chemical Fe oxidation most likely result in the staining issue." in Lines 286–287.

Table 2: It would be helpful to clarify a few things. To avoid any possible misinterpretation, it would help to specify in the caption that RGB refers to separate (R, G, and B) color channels rather than for example a full color RGB light values. Also, some more details should be included in relation to no. 3 and 5 to clarify the difference between these two terms – presumably delta RGB refers to all of the color channels whereas DBI only refers to the blue channel? Or is there some difference when it comes to how delta B (in RGB) was calculated compared to DBI? Unless I am mistaken, the settings for calculating DBI (i.e. how LBI and EBI are determined) in CooRecorder are adjustable – the main settings for this calculation should probably also be stated here.

Response: We clarified each parameter in the Table 2 and its caption (see supplementary reply letter). In short, the main difference between No. 1–3 and No. 4–6 is that No. 1–3 provide one R, G, and B value for each measured lath since all tree-ring measurements of each type (R, G, and B) were averaged by lath. Another difference between No.3 and No.5 is that No.3 is based on differencing R, G, and B intensities measured from 100% of pixels in latewood and earlywood, while DBI, i.e. No.5, is based on differencing LBI (measured from 30% of darkest pixels) and earlywood BI (measured from 100% of the pixels).

Figure 1 caption: 'The gray shades in (a) correspond to . . .' / 'The gray shading in (a) corresponds to . . .'. Also, maybe simply use the term 'replication' instead of 'distribution range' here?

Response: The shading in a) shows the range distribution of black spruce, rather than the replication of trees. We corrected "shade" to "shading" and added the source of the tree distribution map.

Figure 2: Certainly in the main text and maybe also here (e.g. in the figure caption) define MP-AES. For text in right-middle box consider: 'Residual iron content per gram of wood . . .'. Also, it may be more suitable to use past tense in the text boxes: e.g. 'lath1 digested using . . .'; 'lath 2 rinsed in de-ionized water for 2 hrs, then air dried and sanded', etc.

Response: We added the full name of MP-AES in the text (at Line 112) and figure caption. The past tense was used in Figure 2.

Figure 3 caption: The description for (a) should make it clear that these are multiple laths from one LST - for example: '(a) shows examples of differently treated laths from one LST and one living tree sample (last row)'. When referring to panels (b) and (c), it would be clearer to write it using the following format: (b) description, and (c) description – (instead of (b), and (c) description)

Response: We modified the Figure 3 caption: "Figure 3. Residual iron (Fe) and wood RGB intensities (see definitions in Table 2) of LST laths treated with seven chemical reagents. (a) shows examples of treated laths from one LST sample and one living tree sample (last row). The gray outer part of the example LST is discolored due to decay. (b) shows the mean wood RGB intensities according to treatment. (c) shows the mean concentrations of residual Fe according to treatment. Error bars in (b) and (c) refer to standard deviations of corresponding group. Percentages in (c) refer to the Fe removed by de-staining treatments relative to the Fe concentrations of untreated stained LSTs. (d)–(f) show the linear regressions of earlywood, latewood and delta RGB intensities against the log of residual Fe. Regressions are based only on the LST data (circles). Living-tree data are plotted as triangles but are excluded from the regressions.".

Figure 6: Consider also adding some statistics in the panels such as the full period calibration. In the caption, maybe also specify the type of filter used.

Response: Done. See the revised Figure 6 in the supplementary reply letter.

Minor comments: L36: Consider changing to e.g. '. . . high cost of X-ray densitometric equipment' . . . unless this is meant to refer to the relatively high costs associated with processing samples at a 'facility' that is equipped to perform X-ray densitometry. In any case, a small edit is needed.

Response: Corrected.

L37: It may be more accurate to state that the production of BI is 'relatively cheap' rather than 'cheap'.

Response: Corrected to "In contrast, BI is more affordable because it uses commercial flatbed scanners and image analysis software to measure the blue light reflectance of tree rings (Rydval et al., 2014)" in Lines 37–39.

L39: 'coherence' instead of 'coherences'

Response: Corrected.

L43: Perhaps be a bit more specific here by changing 'color issues' for example to something like 'color inconsistencies' or 'inconsistent color properties'?

Response: Corrected to "heterogeneous colors".

L44/45: As this is a property of the wood, 'leading to' is not really appropriate here – instead perhaps ' . . . exhibiting darker heartwood than sapwood'

Response: Yes, that is true. We corrected this sentence to "The best-known issue is the sapwood-heartwood color difference of several tree species such as pine and larch, which does not co-vary with density" in Lines 44–45.

L47: 'occurs' instead of 'happens'

Response: Corrected.

L48: 'LSTs are . . .'?

Response: Corrected.

L49: just 'replication'

Response: Corrected.

L53: The second part of this sentence could be improved for example along the lines of '. . . to realize the potential of the promising BI technique . . .'

Response: We changed this sentence to "Therefore, it is critical to develop unbiased BI data from a variety of wood materials, in particular from the LSTs, to make the promising BI technique widely applicable in future dendroclimatic reconstructions." in Lines 52–53.

L58: 'consists of' instead of 'consists in'; the word 'delta' can be removed from this part of the sentence

Response: Corrected.

L59: Consider changing 'DBI is suitable to recover . . .' for example to 'DBI suitably represents . . .' or 'DBI corrects for . . .'

Response: Corrected.

L64: The last part of the sentence should be re-phrased e.g. '. . . without utilizing low frequency information of the more temperature-sensitive BI data' / '. . . without benefitting from / exploiting any potential improvements in the low frequency domain from the more temperature-sensitive BI data'

Response: Corrected.

L69/70: Consider changing the last part of the sentence to something along the lines of '. . . to a standard comparable to MXD-based reconstructions'

Response: Corrected.

L80/81: minor change needed here, e.g. '. . . LSTs after falling into the water and eventually becoming buried in lake sediments'

Response: Corrected.

L88: 'fungal' rather than 'fungi'? Also, is there a better word that could replace 'invasion' here? Perhaps simply 'fungal discoloration'?

Response: Corrected.

L89: 'before the year'

Response: Corrected.

L94: Should this be 'radius' instead of 'radii' if laths are cut along a single radius, otherwise specify the number of radii

Response: Corrected.

L96: This should probably be 'weighed' instead of 'weighted'

Response: Corrected.

L102: I would recommend presenting the figures in the order that they are first mentioned in the text (Fig. S1 is first mentioned here while Fig. S2 is first mentioned in L131).

Response: We have corrected the order of all figures and tables in the revised manuscript.

L108: This could probably be improved slightly – e.g. 'one lath from each pair of laths'?

Response: Corrected.

L111: Please mention first that MP-AES stands for "microwave plasma-atomic emission spectrometer"

Response: Added at Lines 112.

L112: '. . . weight of the corresponding . . .'

Response: Corrected.

L113: The meaning of this sentence was not immediately clear to me. Consider the following minor revision for the sake of clarity: '. . . represent the combined total . . . cannot be separately distinguished by MP-AES'

Response: Corrected to "Fe concentrations in this study represent the total amount of ferrous and ferric Fe, because MP-AES does not distinguish the type of Fe ions." in Lines 114–115.

L115: 'grit' rather than 'grits'

Response: Corrected.

L118: 'interference' instead of 'inferences'

Response: Corrected.

L122: I think this could be explained a bit better – in a more specific way. Also, 'consistent with' rather than 'consistent to'

Response: We changed the expression to "Because high RGB values represent light colors (i.e. high brightness), they were subtracted from a value of 256 such that smaller RGB values are associated with lighter colors" in Lines 122–123.

L141: perhaps '(i.e. each lath pair)'?

Response: Corrected.

L143: 'An age-dependent spline . . .'

Response: Corrected.

L148/L150: Also in relation to Fig. S8 / S7, see earlier comment about sequential order of figures mentioned above.

Response: We have corrected the order of all figures and tables in the revised

manuscript.

L151: maybe 'poor (tree) health' would be better than 'unhealthy tree growth'

Response: Corrected.

L154: 'averaged into'? Also, maybe consider showing the individual MXD chronologies in the SI.

Response: Added to SI material, Fig. S9. See Fig. S9 in the supplementary reply letter.

L161: This could use a bit of re-wording.

Response: We changed this sentence to "The reconstructions were based on the scaling method (Esper et al., 2005; Rydval et al., 2017) by adjusting means and standard deviations of the chronologies to those of the temperature target over the 1901–2015 time interval" in Lines 161–163.

L165: maybe slightly re-word – e.g. '. . . in order to assess the role of Fe in the staining issue'

Response: We changed this to "in order to assess the roles of Fe in the staining issue" in Line 166–167.

L167: Perhaps just briefly specify the advantage of applying this specific filter in the context of high-pass / low-pass filtering.

Response: We used this method because it is the default option of the dplR package, not because of a specific advantage. The filtered series were very similar between using the Butterworth filter and Chebyshev filter.

L168: 'Performance of the reconstructions was assessed ...'

Response: Corrected.

L170: '. . . while the 1961-2015 period . . .'

Response: Corrected.

L178-179: Just as a general point, I wonder what the reason for this might be? Could it be related to the 'color properties' of the staining caused by Fe?

Response: Yes, this is correct. This is related to the properties of Fe oxides that stained the wood. In general, Fe oxides have stronger red light reflectance. From this aspect, our conclusions are also consistent with this general phenomenon. We feel slightly confused by the real-world colors of the stain on wood, which look more blue than red, however such "blue" stain reflect more red color according to our analysis (Fig. 3b).

L182: 'intensities'

Response: Corrected.

L186-187: '. . . of treated earlywood and latewood'?

Response: Corrected.

L189: Please be more specific here and elaborate on what the difference is.

Response: We changed it to "Note that wood delta BI and DBI were not calculated exactly in the same way (see Table2; delta BI is the averaged difference between BI of entire latewood and earlywood from all tree rings in a sample, while DBI is a tree-ring parameter which presents the difference between LBI and BI of entire earlywood for each tree ring)." in Lines 194–196.

L191: Consider something along the lines of '. . . four treatments examined in more detail . . .'

Response: Sorry, but we could not understand this comment

L192: 'prior to 1900 CE' or 'prior to the year 1900 CE'

Response: Corrected.

L193: 'coherence' rather than 'coherences'

Response: Corrected.

L196: 'few differences' instead of 'little differences'

Response: We changed the sentence to "In addition, few differences were found between the control DBI series and chemically treated DBI data, although the colors of wood samples were visually distinct (Fig. 3a)." in Lines 203–204.

L203: 'Briefly' is probably not needed here

Response: We prefer to keep it. The term is suggested by one of the co-author working in geochemistry, because the chemistry of iron is complex.

L208: I would recommend re-wording 'combine to wood' – 'combine with wood' or maybe 'bind to wood'?

Response: We corrected it to "bind".

L225: just 'etc.' instead of 'and etc.'

Response: Corrected.

L228: maybe 'in our samples' rather than 'from our samples'?

Response: Corrected.

L229-230: Minor edit needed here: e.g. 'is not sensitive to sulfur and phosphorous' / 'is not designed to detect sulfur and phosphorous' or 'is not sufficiently sensitive to detect . . .'?

Response: We made some changes to clarify this. "However, the MP-AES instrument is not sufficiently sensitive to verify our hypothesis regarding those Fe complexes (detection limits are ∼6500 ppb and 125 ppb for dissolved sulfur and phosphorus, respectively, compared to ∼4.6 ppb for dissolved Fe)" in Lines 236–238.

L234: '. . . very little residual Fe . . .'

Response: Corrected.

L236: probably replace 'such' with 'the'

Response: Corrected.

L240: maybe something like '. . . when staining is present in subfossil wood' would be better

Response: Corrected.

L242: 'a' should be removed

Response: Corrected.

L247: something is missing here – e.g. '. . . for example as with the most efficient MixC protocol' or something similar

Response: Corrected.

L250-251: 'This evidence suggests that DBI is not only an excellent solution to resolving ...'

Response: Corrected.

L251: 'but also efficiently resolves ...'

Response: Corrected.

L256-257: The last part of this sentence could be re-worded and expanded a bit to clarify what is meant by this.

Response: We made some corrections. "Although DBI is theoretically sufficient to solve the sapwood-heartwood color issue (Björklund et al., 2014), in our case it could only partially correct this problem (Fig. S8). Old living trees were collected from lakeshore forests at the L20 site and they often displayed declining ring widths compared to healthy trees sampled later at the same site (not shown). DBI of L20 is likely

influenced by these narrow tree rings (Björklund et al., 2019) because DBI of black spruce is not only correlated to MXD but also to the ring-width data (Wang et al., submitted). We thus speculate the divergence of DBI reflects mostly a specific issue related to the declining growth of unhealthy trees." in Lines 262–267.

L258: replace 'declined' with 'declining'

Response: Corrected.

L259: 'decline' instead of 'declines'? Also, the reference Björklund et al., 2019 (Reviews of Geophysics) may also be relevant to this point.

Response: Corrected and reference added.

L261: 'need to be' rather than 'need be'?

Response: Corrected.

L268-269: '. . . further attention / investigation is needed . . .'

Response: Corrected.

L281: should probably be '. . . used as part of Fe extraction protocols . . .'

Response: Corrected.

L282: 'which also face / are also susceptible to / also suffer from'?

Response: Corrected.

Please also note the supplement to this comment:
https://www.biogeosciences-discuss.net/bg-2020-102/bg-2020-102-AC1-supplement.pdf

**Supplement:**

**Supplement reply letter to Meloš Rydval**

[Figure]

**Revised Figure 6. Temperature reconstructions using the LBI chronology for the MixC protocol (purple), the Control DBI chronology (blue), and the reference MXD chronology (red) for the 1655–2015 (a, c) and 1901–2015 (b, d) time intervals. (e) shows the 1-year-lag moving EPS computed in 31-year windows. Thick smooth lines denote the 10-year low-pass series filtered using the Butterworth filter. Vertical gray bars show the periods where tree replication is less than 10 (Fig. S3c).**

[Figure]

**Figure S9. MXD regional chronologies for the MixA, MixB, MixC, and Control treatments.**

**Revised Table 2. Definitions of wood color intensities and tree-ring parameters used in this study. The RGB intensities refer to three color intensities measured separately from red (R), green (G), and blue (B) channels. The parameters No.1–3 are used to quantify wood colors while the parameters No. 4–6 are conventional dendrochronological parameters.**

| No. | Parameter | Definition |
|---|---|---|
| 1 | earlywood & latewood RGB intensities* | Mean R, G and B intensities from all pixels of earlywood or latewood (see Fig. S6), averaged from all tree rings of each wood lath. |
| 2 | wood RGB intensities* | Mean RGB intensities averaged from earlywood and latewood RGB intensities. |
| 3 | delta RGB intensities | Earlywood RGB intensities subtracted from corresponding latewood RGB intensities |
| 4 | LBI* | Mean blue intensity of 30% of the darkest pixels in the latewood of each tree ring (Fig. S6). |
| 5 | DBI | Raw LBI (measured from 30% of the darkest pixels) subtracted from raw earlywood BI (measured from 100% of pixels), automatically derived from CooRecorder 8.0 for each tree ring. |
| 6 | MXD | The maximum value of measured tree-ring latewood density. |

*: data were inverted by subtracting the raw data from a value of 256.

---

## Author Comment (AC2) · 14 Jun 2020

We are grateful for the useful comments from Dr. Jesper Björklund. Our responses are listed below:

The manuscript by Wang et al presents a very interesting sample material for temperature reconstructions and examine how to best utilize this in conjunction with the popular and affordable BI technique. The paper is foremost dedicated to a very novel and clever de-staining experiment which I thoroughly enjoyed and have the potential to be highly cited in future BI studies. The second component was a careful comparison of LBI, DBI and MXD from parallel X-ray measurements to evaluate the performance

of the chemical de-staining and LBI and DBI parameters with MXD as reference. Although the authors conclude that the simple DBI was more successful in replicating the low-frequency variance of the MXD, they have made some very important discoveries in terms of de-staining of relict wood material. The DBI parameter appears to be quite successful, but has documented problems as the authors also mention in the final sentences. Therefore, all tools available for de-staining prior to DBI transformation must be considered of great value. I congratulate the authors to a fine, and from what I can tell labor intensive, experiment and I consider the manuscript suitable for publication following minor revisions and clarifications. I also look forward to learn more about the planned follow-up manuscript.

Detailed comments: L32 We would not say BI is recently developed anymore, it has been around almost 20 years now.

Response: Yes, we totally agree. We removed "recently developed".

L32-33 The BI technique is an alternative to the X-ray technique in producing proxy parameters such as MXD.

Response: We re-phrased the first sentence as: "The blue intensity (BI) technique is an alternative to the more expensive X-ray densitometric methodology in producing tree-ring proxy parameters such as maximum latewood density (MXD) for dendroclimatology." in Lines 32–33.

L37 -38 Consider changing to something like: "In contrast, BI is more affordable because of the utilization of commercial flatbed scanners to generate images of reflected blue light analyzed in potentially affordable image analysis software. . ."

Response: We re-phrased this sentence as "In contrast, BI is more affordable because it uses commercial flatbed scanners and image analysis software to measure the blue light reflectance of tree rings." in Lines 37–39.

L38-42 Strange sentence, some of the studies encouraging more studies were made

later than the encouraged studies. Work a bit more on this sentence and consider also these references: Björklund et al., 2014, 2015; Dolgova, 2016; Fuentes et al., 2018; Kaczka et al., 2017; McCarroll et al., 2013; Rydval, Gunnarson, et al., 2017.

Response: Several of the suggested references are cited elsewhere in the manuscript. Here we only wish to cite studies specific to latewood BI rather than delta BI, because the topic of delta BI is detailed in the next paragraph. So we only added McCarrol et al., 2013. We improved the sentence to make it more fluent. "Excellent coherence was reported between the latewood BI (LBI) and MXD data measured from living-tree materials of a number of coniferous tree species across the northern hemisphere (Campbell et al., 2007; Kaczka et al., 2018; Österreicher et al., 2015; Rydval et al., 2014; Wilson et al., 2014), suggesting the potentials to use BI method in dendrocliamtic reconstructions (McCarrol et al., 2013; Rydval et al., 2017; Wilson et al., 2019)." in Lines 39–42. The later three references (McCarrol et al., 2013; Rydval et al., 2017; Wilson et al., 2019) are real reconstruction works.

L45 Should perhaps add something like: "..not accompanied by a similar difference in density.."

Response: I corrected this sentence to: "The best-known issue is the sapwood-heartwood color difference of several tree species such as pine and larch, which does not co-vary with density." in Lines 44–45.

L83 newly exploited lake?

Response: This has been changed to "newly sampled lake".

L84 millennium-long?

Response: We think the meanings of "millennium-long" and "millennial" are very similar. We keep "millennial" here.

L96 What was the purpose of the weighing? Were the laths also weighed after the chemical analysis? Could not find any more use of these measurements in the

manuscript

Response: The weights of wood laths were used to calculate the Fe concentrations in the wood. This is because Fe concentrations in wood depend strongly on the weights of wood laths. The iron data shown in the original manuscript had already been adjusted by the wood weights. It was described at Line 113 of the revised manuscript, "data were adjusted to "milligram of Fe per gram of wood" according to the dilution and weight of the corresponding wood lath.". We only weighed a few post-treatment wood laths. This is because we set two control groups in addition to the chemical treatments, i.e. untreated stained samples and living tree samples. For these samples, we would not know their post-treatment weights. A consistent measure of weight from laths should be used, i.e. pre-treatment weight. Second, the amount of wood burrs contacted on the surface of laths might also affect the post-treatment weights. These burrs were often found detached from the laths after treatments (samples are placed in a tube on a shaker for at least 24 hrs). This mass loss varies tree by tree, and we cannot quantify the weight losses. For example, being similarly treated with MixC, one lath lost 0.2% of weight relative to a pre-treatment weight (0.2866g) while another lath could loss 1.03% of its weight (pre-treatment weight was 0.1935g).

L104 .., to identify the most effective. . .? Remove "(see results below)". The results are always be presented after the methods description.

Response: Ok, we removed "(see results below)".

L118 sensu Rydval et al., 2014?

Response: We added this reference here.

L118-119 Great initiative L121-122 Very strange statement. Real world observations? Do you mean: lower RGB values corresponds to lighter densities?

Response: We changed the expression to "Because high RGB values represent light colors (i.e. high brightness), they were subtracted from a value of 256 such that smaller

RGB values are associated with lighter colors." in Lines 122–124.

Section 2.2, 2.3 and 2.4 Consider re-structuring here. Perhaps one section for chemical de-staining description. One section for BI and X-ray data development and one section for chronology development for climate analysis, and sample average RGB data?

Response: The manuscript comprises a de-staining experiment and a dendroclimatic assessment. Accordingly, we kept section 2.2 and merged sections 2.3, 2.4 and 2.5 to obtain one section for the de-stanning experiment (section 2.2) and one section for the dendroclimatic assessment (section 2.3)

L139 Did you use the full RGB spectrum or only the blue spectrum? If the latter, it is consistent with the use of BI based parameters. Same comment in L165.

Response: We extracted the red, green and blue intensity values from earlywood and latewood of each ring, then averaged values by colors and laths. Of course, for the LBI and DBI we considered only the blue spectrum. A revised version of Table 2 (in the supplementary reply letter) clarifies the measured parameters. We could not find the comment in L165.

L145 N.B. residuals are most often used for density related parameters. This is not a major problem here since you compare results from BI and X-ray, but may be important in pure climate reconstructions.

Response: Ok, we will think of calculating residuals in future reconstructions.

L168 "coherence" can also be a type of statistical analysis, perhaps change to the more general term of "agreement", or simply not explain correlation since more or less the entire readership is familiar with this.

Response: We changed it to "agreement".

Figure S4 Spelling of replication

[Figure]

Response: Corrected.

Figure s6 spelling of earlywood. It seems odd that the area of the 30% of the darkest pixels in the latewood are differently sized even though the latewood area is roughly the same (compare ring 4 and ring 5). Please check the definition you used and clarify why this is the case.

Response: Misspelling Corrected. We systematically used the 30% of the darkest latewood BI as LBI. However, the Figure S6b was generated using the densitometer function of CooRecorder (See Figure1 in the supplementary reply letter). We replaced this figure with the output of the actual LBI measurements (Figure2 in the supplementary reply letter).

L169-171 Would be great to have running Rbar or EPS, to evaluate the difference between the different parameters. Perhaps this can explain why the DBI perform so badly in the post 1960 period compared to LBI and MXD. Both in terms of trend and correlation.

Response: We added EPS values in the revised Figure 6 (in the supplementary reply letter). Some relevant discussions were made on the EPS data (at Line 273 in the revised manuscript). It is true that EPS of DBI was more unstable than LBI.

L182 spelling intensities

Response: Corrected.

Figs. S7-S8 Would be interesting to also present the Earlywood measurements. Would be even more interesting if you also presented Delta density and Earlywood density. It is puzzling why LBI and DBI has such similar trends in S7. Is there a HW/SW transition in these trees, if so why so weak in the earlywood? Are the rings in the post 1960 period very narrow? If so, I think that your measurement resolution is causing some problems here. Consider that the measurement resolution is affecting your latewood measurements more than your earlywood measurements. That is, your latewood BI

is deflated because of adjacent contamination of earlywood BI. Ergo the delta BI will be artificially lowered and similar in trend to LBI. Not completely relevant to your nice study, but could not resist :)

Response: We did not extract the earlywood BI and earlywood density data for this manuscript, which are in very raw forms, and consequently delta density data are not available. We responded to the rest of this comment below, under the L253–260 comment.

L196 check grammar

Response: We changed the sentence to "In addition, few differences were found between the untreated control DBI series and chemically treated DBI data, although colors of wood samples were visually distinct (Fig. 3a)" in Lines 204–205.

L208 combine to wood? Not clear, rephrase..

Response: We replaced "combine" with "bind".

L241-242 This is not surprising. If you would calculate delta density and correlate with delta BI you would probably find equally high correlation as between LBI and MXD. This is not needed in revision, I am merely pointing this out.

Response: OK. We deleted this sentence.

L253-260 I think you are right that the narrow ring widths are causing the problem here, but I would not say it is a healthy versus unhealthy tree problem. It is a problem of measurement resolution (see comment above for fig s7). Healthy tree can also have narrow rings.

Response: Yes. The L20 samples showed narrow rings and distinct sapwood-heartwood colors. This latter phenomenon is unusual for black spruce because this species usually does not show distinct boundary between earlywood and latewood. We speculate that this phenomenon was resulted from the unhealthy state of some

trees in this site because in the early stages of sampling, we collected trees as old as possible, regardless of their crown shape and growth rate. We agree that the cause of the divergence (in particular for the DBI which in theory is not affected by color) is most likely the narrow rings. We thus modified Lines 155–160 in the original manuscript to:

"Old living trees were collected from lakeshore forests at the L20 site and they often displayed declining ring widths compared to healthy trees sampled later at the same site (not shown). DBI of L20 is likely influenced by these narrow tree rings (Björklund et al., 2019) because DBI of black spruce is not only correlated to MXD but also to the ring-width data (Wang et al., submitted). We thus speculate the divergence of DBI reflects mostly a specific issue related to the declining growth of unhealthy trees." in Lines 263–267.

L262 yes interesting observation. Would be better underpinned if you also presented the rbar for all the parameters.

Response: We added moving EPS values in Figure 6 (see revised Fig. 6 in supplementary reply letter).

Please also note the supplement to this comment:
https://www.biogeosciences-discuss.net/bg-2020-102/bg-2020-102-AC2-supplement.pdf

**Supplement:**

**Supplement reply letter to Jesper Björklund**

[Figure]

**Figure 1. The output of the densitometer function of Correcorder 8.0 used to generate Figure S6b in the original manuscript.**

[Figure]

**Figure 2. The new Figure S6**

[Figure]

**Revised Figure 6. Temperature reconstructions using the LBI chronology for the MixC protocol (purple), the Control DBI chronology (blue), and the reference MXD chronology (red) for the 1655–2015 (a, c) and 1901–2015 (b, d) time intervals. (e) shows the 1-year-lag moving EPS computed in 31-year windows. Thick smooth lines denote the 10-year low-pass series filtered using the Butterworth filter. Vertical gray bars show the periods where tree replication is less than 10 (Fig. S3c).**

**Revised Table 2. Definitions of wood color intensities and tree-ring parameters used in this study. The RGB intensities refer to three color intensities measured separately from red (R), green (G), and blue (B) channels. The parameters No.1–3 are used to quantify wood colors while the parameters No. 4–6 are conventional dendrochronological parameters.**

| No. | Parameter | Definition |
|---|---|---|
| 1 | earlywood & latewood RGB intensities* | Mean R, G and B intensities from all pixels of earlywood or latewood (see Fig. S6), averaged from all tree rings of each wood lath. |
| 2 | wood RGB intensities* | Mean RGB intensities averaged from earlywood and latewood RGB intensities. |
| 3 | delta RGB intensities | Earlywood RGB intensities subtracted from corresponding latewood RGB intensities |
| 4 | LBI* | Mean blue intensity of 30% of the darkest pixels in the latewood of each tree ring (Fig. S6). |
| 5 | DBI | Raw LBI (measured from 30% of the darkest pixels) subtracted from raw earlywood BI (measured from 100% of pixels), automatically derived from CooRecorder 8.0 for each tree ring. |
| 6 | MXD | The maximum value of measured tree-ring latewood density. |

*: data were inverted by subtracting the raw data from a value of 256.